# Do They Look Where They Go? Gaze Classification During Walking

**Gianni Bremer**                                     GIANNI.BREMER@WWU.DE
**Niklas Stein**                                       NIKLAS.STEIN@WWU.DE
**and Markus Lappe**                               MLAPPE@UNI-MUENSTER.DE
*Department of Psychology*
*University of Muenster*
*48149 Muenster, Germany*

**Editor:** Editor's name

## Abstract

In many applications of human-computer interaction a prediction of the human's next intended action is highly valuable. For locomotor actions, in order to control direction and orientation of the body a walking person relies on visual input obtained by eye and head movements. The analysis of these parameters can be used to infer the intended goal of the walker. However, such a prediction of human locomotion intentions is a challenging task since interactions between these parameters are non-linear and highly dynamic. Distinguishing gazes on future waypoints from other gazes can be a helpful source of information. We employed LSTM models to investigate if gaze and walk data can be used to predict whether walkers are currently looking at locations along their future path or whether they are looking in a direction that is away from their future path. Our models were trained on egocentric data from a virtual reality experiment in which 18 participants walked freely through a virtual environment while performing various tasks (walking along a curved path, avoiding obstacles and searching for a target). The dataset included only egocentric features (position, orientation and gaze) and no information about the environment. These features were used to determine when gaze was directed at future waypoints and when not. The trained model achieved an overall accuracy of 80%. Biasing the model to focus on correct classification of gazes away from the path increased the detection rate of these gazes to 90%. An analysis of model performance in the different walking task showed that accuracy was highest (85%) for curved path walking and lowest (73%) for the target search task. We conclude that online gaze measurements during walking can be used to estimate a walker's intention and to determine whether they look at the target of their future trajectory or away from it.

**Keywords:** LSTM; Virtual Reality; Eye Tracking; Locomotion; Path prediction; Machine Learning; Gaze

## 1. Introduction

Where we look is tightly linked to our actions as we use eye and head movements to collect visual information for action control (Land and Tatler, 2009). Thus, eye movements typically precede other motor actions (Land and Hayhoe, 2001; Hayhoe and Ballard, 2005). Therefore, gaze signals can be highly valuable for predicting future actions (Belardinelli et al., 2016; Gandrud and Interrante, 2016; Zank and Kunz, 2016a,b). Specifically, during

locomotion walkers often direct their gaze towards their target at some point before approaching it (Hollands et al., 2002; Durant and Zanker, 2020). Yet, gaze during walking is also directed to obstacles, or to the ground a front when walking in in uneven terrain (Hollands et al., 1995; Hollands and Marple-Horvat, 1996; Calow and Lappe, 2008; 't Hart and Einhauser, 2012; Matthis et al., 2018), or towards the inside of a curve when walking along a curved path (Grasso et al., 1998; Imai et al., 2001). Moreover, eye movements are linked to changes in walking direction (Hollands et al., 2002) and to searching for targets between distractors (Kit et al., 2014). Therefore, while gaze contains information about the intentions of a walker it does not immediately identify the target of a walk.

Yet, the relationships between walking and gazing still allows to predict where one is going (Zank and Kunz, 2016a; Gandrud and Interrante, 2016; Cho et al., 2018; Bremer et al., 2021; Stein et al., 2022). Recent work (Bremer et al., 2021; Stein et al., 2022) employed machine learning models to investigate if walk and gaze data can be used to predict the future location that a walker will attain. They collected training data for the models in a virtual reality experiment in which 18 participants walked freely through a virtual environment while performing various tasks (walking in a curve, avoiding obstacles and searching for a target). The recorded position, orientation- and eye-tracking data were used to train an LSTM model (Hochreiter and Schmidhuber, 1997) to predict the position of the walker 2.5 seconds into the future. The best model predicted free walking paths with a mean error of 66 cm. Comparing the influence of different features on the prediction quality showed that gaze and orientation of the head and body provided significant contributions. Even a model using only gaze data was able to predict the future position with an error of only 78cm. Comparing different behaviours indicated that gaze offered the greatest predictive utility in situations in which participants were walking short distances or in which participants changed their walking speed.

In the present work we trained an LSTM model to classify whether any particular gaze during a walk is directed towards a location along the future path or rather towards a location in the environment that is not on the future path, like, for example, an obstacle. This classification might be used to further improve prediction of a walkers future path but it could be also useful for many other applications in human-computer, human-environment, or human-human interactions that involve natural walking.

## 2. Method

The data to train the model was taken from a dataset in which 18 participants (8 female, 10 male) completed a set of natural locomotion tasks in virtual reality (Bremer et al., 2021), freely available from https://osf.io/b43uv/. It contained free walking data in environments and tasks designed to include typical behaviors, such as searching for a target object, walking along a curve and avoiding obstacles. The virtual environment consisted of two rooms linked by a corridor. The first room contained a target object, which the participant had to search for. This target was placed among six identical looking distractor objects (see Figure 1). Participants had to perform a search by walking freely between the objects and inspecting them closely until they found the target. In the second room, the aim was to walk to a target while avoiding an obstacle in the middle between the start location and the target. The two rooms were connected by a transition corridor that followed a

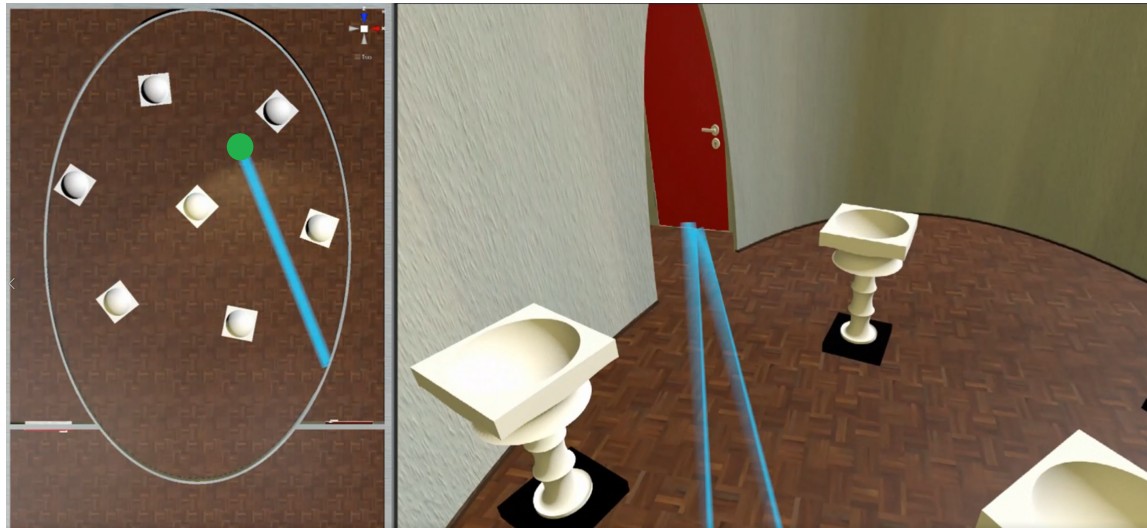

Figure 1: Left: Top view of the search room of the virtual environment containing seven search objects. The user is shown as a green circle. The blue lines show the user's gaze direction. Right: Ego perspective. The user looks at his current walking target, the door that connects to the transition corridor.

curve with a radius of 5.5 m. Participants completed a total of 10 trials in each room, going back and forth between the rooms in between. The virtual environment was presented on an HTC Vive Pro Eye head-mounted display (HMD) with a resolution of 1440×1600 pixels per eye, a frame rate of 90 Hz and a field of view of 110 degrees. The HMD provided head position and orientation tracking as well as eye gaze direction at 90Hz sampling rate. A Vive tracker attached to the back measured body orientation.

## 2.1. Preparation

Data was divided into 50-millisecond bins each corresponding to about four frames in the raw data. To form the models' inputs, sequences containing the data at the current timestamp (the time at which the prediction is calculated) and the data of some immediately preceding timestamps were then constructed. The length of the input was set to 2.5 seconds (50 samples per input). To compensate for potential spatial asymmetries every second sequence was mirrored on the XZ-plane.

To deal with blinks, a single missing value in the eye-tracking data was filled using linear extrapolation based on the previous 3 frames. Data sequences with multiple subsequently missing values were excluded. Additionally, data containing prolonged standing (e.g. at the beginning of the experiment) in the HMD tracking data were excluded using a threshold of 0.15 m/s.

Features for training the model consisted of: the 2D horizontal velocity of the head in the space, the orientation (yaw and pitch) of the head, the yaw angle between body and head, and the yaw and pitch of the gaze direction with respect to the head. Labels

were built by marking if the gaze direction at each time point was targeted at any point along the subsequent walking trajectory according to the following criteria: The trajectory was a segment of the path that the user walked between 0.5 to 2.5 seconds in the future. Horizontal gaze had to be between -5 and +5 degrees from this trajectory. Vertical gaze had to be at eye level or downward, i.e., gazes towards the sky were excluded.

## 2.2. Prediction Model

Our LSTM model had three layers of 64 hidden units each. After each LSTM layer a dropout layer (p = 0.2) was implemented (Srivastava et al., 2014) resulting in the final linear dense layer with one output, that was transformed with a sigmoid function to form the model output that could be rounded. For training, the sigmoid function was combined with a binary cross entropy loss function into one layer. We used adam as the optimizer (Kingma and Ba, 2014).Learning rate was set to 0.001. The model was trained for 50 epochs using a batch size of 64. Then the epoch with the lowest validation error was selected.

To avoid overlapping input sequences in the training and test set and to ensure the transferability of a model to new data, cross-validation was implemented at group level. In this process, leave-3-out-cross-validation was used. Before training, features and labels were z-standardized. To fit the scalers, only the training set was used while all data was adjusted with these scalers.

In addition to this model, we also report a model that was biased by shifting the threshold of the sigmoid function to increase the detection of gazes away from the future trajectory at the expense of the detection rate for gazes at waypoints.

## 3. Results

The 50-sample input sequences and prediction labels formed 115,533 input-output pairs in total. In these data, 44.6% of gazes were directed to a future waypoint. The model was able to distinguish gazes to future waypoints from other gazes with 80.2% accuracy. When differentiating between type I and type II errors we found that true gazes to a walking target could be detected by the model with 86.4% accuracy, whereas gazes to other points were detected with 73.8% accuracy (Figure 2).

Depending on the application context, one could envision situations in which it is preferable to accept a high false negative rate in order to reduce the false positive rate. Specifically, for applications such as steering or attention control it might be most important to reliably detect situations in which the user looks away from their future path. A high false positive rate is acceptable also because a typical fixation of 300 ms duration consists of several data points at 50ms bins. Therefore it might be sufficient if some points of a fixation are recognized correctly, while other points are missed. Therefore, we also report a model with bias shifted to filter gazes not directed to future waypoints. The bias (intercept) was adjusted to produce a false negative rate of 50%. This decreases the general performance to 73.1%, but gazes that were not aligned to the trajectory are correctly classified at 89.6% (Figure 2).

Gaze behavior during walking is linked to the intention and task of the walker and differs between walking amongst obstacles, walking towards a target or walking along a curved path(Hollands et al., 1995; Hollands and Marple-Horvat, 1996; Grasso et al., 1998; Imai et al., 2001; Hollands et al., 2002; Calow and Lappe, 2008; 't Hart and Einhauser, 2012;

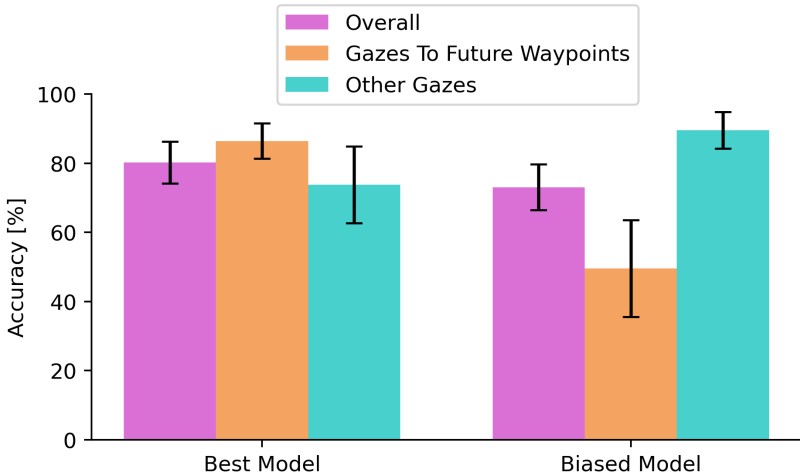

Figure 2: Accuracy of the classification by the models. Left: unbiased model. Right: model biased to produce better accuracy for detecting gazes that are not on a future waypoint at the cost of less accuracy for gazes along the way. In each case overall accuracy is reported as well as detection accuracy for each of the two classes.

Matthis et al., 2018; Durant and Zanker, 2020). This leads to different levels of information that is contained in gaze in these situations. The dataset we used contained several different tasks in different rooms(Bremer et al., 2021; Stein et al., 2022). We, therefore, also compared accuracy between the different rooms for the unbiased model. For the free exploration task in the search room the model reached 72.6% accuracy. In the transition corridor, where the focus was on curves, the model reached an accuracy of 84.9%. The analysis of the obstacle room resulted in a 79.0% prediction accuracy.

## 4. Discussion

We presented a model to classify current egocentric gaze data from freely walking subjects into on-path and off-path gazes based on the immediate feature history of the walker's position, orientation and gaze. This method could be used as a preprocessing step for locomotion prediction models but also for various applications in which gaze data needs to be classified or interpreted for other purposes. These include, for example, human-computer interfaces involving walking in extended reality or controlling assistive robots, pedestrian walking prediction, e.g., for self-driving cars, or monitoring of attention, e.g., in the elderly to reduce risk of falling.

All features we used were egocentric features, i.e. the walker's locomotion and orientation of the body and eyes. Thus, our model has no knowledge of the environment. While the walking and gazing behavior surely depends on environmental factors we aimed to produce a system that can classify gazes in any environment in an invariant manner. However, a limitation lies in the constrained nature of our data set. Although it combines several rooms and tasks we cannot rule out that movements in other types of tasks are underrepresented. Nevertheless, we believe that this system can be generalized to some extent to other virtual

and real environments. Future research should aim to use different data sets in diverse environments in which the input features can be obtained.

While we believe that the concentration of only egocentric features is a strength of our model, additional data, if available, might be used to increase the performance. This does not need to be full environmental information (e.g. visual or range image depth maps and identifiable obstacles), also simpler and more transferable, camera-based data such as optic flow could be useful, since eye movements during walking also depend systematically on optic flow (Lappe et al., 1998; Niemann et al., 1999; Knöll et al., 2018; Chow et al., 2021). Also brain data, for example from EEG recordings could be helpful.

## 5. Conclusion

We reported on deep learning classification of a walker's gaze alignment with their future path using a time-sequence of prior position, orientation, and gaze data. We conclude that a model using the LSTM architecture can be used to distinguish gazes directed at future waypoints from gazes to other points. Moreover, our results suggest that the model can be modified to specifically detect gazes that are not targeted at future positions

## Acknowledgments

This work was supported by the German Research Foundation (DFG La 952-4-3, La 952-7) and has received funding from the European Union's Horizon 2020 research and innovation programme under grant agreement No 951910. The authors declared no potential conflicts of interest with respect to the research, authorship, and/or publication of this article.

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
