# OpenReview forum: "Do They Look Where They Go? Gaze Classification During Walking"
_NeurIPS.cc/2022/Workshop/GMML — Gaze Meets ML 2022 Poster_

### Official Review · Reviewer_SX1J · 2022-10-17
**Interesting work**

**Rating:** 6
**Confidence:** 4

**Review:**

The authors evaluate the accuracy of gaze tracking in a virtual environment as it relates to the hypothesis that we look where we walk. So, if we can estimate gaze, we can estimate alignment with target. The method just uses position, orientation, and gaze data to estimate agreement with the path.

The work that is described is relatively easy to read and the results are fairly good.

I do have an issue with the hypothesis that the environment does not matter in path gaze focus. In familiar environments, we could gaze off the path more frequently than in unfamiliar environments. This has not been modeled. The pace used to move also affects the focus. Debris in the path affects gaze.

So, the problem has been simplified to a point where it is nice but insufficient.

---

### Official Review · Reviewer_wn6K · 2022-10-19
**Good paper, well written, interesting findings**

**Rating:** 7
**Confidence:** 3

**Review:**

The authors train a model that predicts on-path and off-path gazes.

The authors provide detailed descriptions of the data and the model hyperparameters.

The paper is well written, and easy to follow, and the task is useful to the community.

---

### Meta-Review · Area_Chair_on9i · 2022-10-20

**Recommendation:** Accept (Poster)
**Confidence:** 5

**Metareview:**

The authors present a methodology on using eye tracking in a virtual environment. The main proposed idea is to "distinguish gazes directed at future waypoints from gazes to other points".  The idea is very interesting as it can have appealing applications to visual attention search.

---

### Decision · Program_Chairs · 2022-10-20

Accept (Poster)